# What change in body mass index is associated with improvement in percentage body fat in childhood obesity? A meta-regression

Laura Birch,[1] Rachel Perry,[1] Linda P Hunt,[1] Rhys Matson,[1] Amanda Chong,[1] Rhona Beynon,[1] Julian PH Shield[1,2]

[1]NIHR Bristol Biomedical Research Centre—Nutrition theme, Level 3 University Hospitals Bristol Education Centre, Bristol, UK
[2]University Hospitals Bristol NHS Foundation Trust, Level 6 University Hospitals Bristol Education Centre, Bristol, UK

**Correspondence to**
Laura Birch;
laura.birch@bristol.ac.uk

## ABSTRACT

**Objective** Using meta-regression this paper sets out the minimum change in body mass index-SD score (BMI-SDS) required to improve adiposity as percentage body fat for children and adolescents with obesity.

**Design** Meta-regression.

**Setting** Studies were identified as part of a large-scale systematic review of the following electronic databases: AMED, Embase, MEDLINE via OVID, Web of Science and CENTRAL via Cochrane library.

**Participants** Individuals aged 4–19 years with a diagnosis of obesity according to defined BMI thresholds.

**Interventions** Studies of lifestyle treatment interventions that included dietary, physical activity and/or behavioural components with the objective of reducing obesity were included. Interventions of <2 weeks duration and those that involved surgical and/or pharmacological components (eg, bariatric surgery, drug therapy) were excluded.

**Primary and secondary outcome measures** To be included in the review, studies had to report baseline and post-intervention BMI-SDS or change measurements (primary outcome measures) plus one or more of the following markers of metabolic health (secondary outcome measures): adiposity measures other than BMI; blood pressure; glucose; inflammation; insulin sensitivity/resistance; lipid profile; liver function. This paper focuses on adiposity measures only. Further papers in this series will report on other outcome measures.

**Results** This paper explores the potential impact of BMI-SDS reduction in terms of change in percentage body fat. Thirty-nine studies reporting change in mean percentage body fat were analysed. Meta-regression demonstrated that reduction of at least 0.6 in mean BMI-SDS ensured a mean reduction of percentage body fat mass, in the sense that the associated 95% prediction interval for change in mean percentage body fat was wholly negative.

**Conclusions** Interventions demonstrating reductions of 0.6 BMI-SDS might be termed successful in reducing adiposity, a key purpose of weight management interventions.

**Trial registration number** CRD42016025317.

## Strengths and limitations of this study

► We believe that this is the first paper to attempt to bring together all studies that have reported both a change in body mass index-SD score and changes in a marker of adiposity in the paediatric population with obesity.

► The systematic methods employed to identify the included studies were stringent, but it is possible that some relevant studies might have been missed.

► There was some variation in the reporting of results where there were multiple publications of the same study; in these cases, the results from the most comprehensive paper have been used.

► Studies that did not report change in mean percentage body fat could not be included in this meta-regression.

## INTRODUCTION

Childhood obesity is one of the most serious global public health challenges of the 21st century.[1] In England, the latest figures from the National Child Measurement Programme, which measures the height and weight of around 1 million school children every year, showed that 9.5% of children aged 4–5 years and 20.1% of those aged 10–11 years were obese.[2 3] Childhood obesity has adverse health consequences in both the short-term and long-term, including an increased risk of developing metabolic disturbances, like hypertension, dyslipidaemia and insulin resistance, and becoming obese adults.[4] The presence of adverse changes in cardiac and vascular function and type 2 diabetes, which were previously considered adult morbidities, now being identified in children and adolescents with obesity[5–11] illustrates the urgent need for effective weight management treatment interventions to reduce adiposity and improve the metabolic health status of the paediatric population.

Moderate weight loss has been shown to have a positive impact on many metabolic and cardiovascular risk factors.[12 13] Weight management interventions for adults with

obesity that result in a 5–10% decrease in body weight are associated with significant improvements in blood pressure, serum lipid levels and glucose tolerance[14] and reduction in the prevalence of hypertension and diabetes.[15] Minimum weight management targets can therefore be set to improve metabolic health in this population.[16]

During childhood, all measurements over time are complicated by the influence of growth, meaning that cut-offs routinely used in the adult population cannot be used in children and adolescents. However, measured values of body mass index (BMI) can be standardised into SD scores (SDS) with respect to reference populations.[17] These standardised scores, referred to as BMI-SDS throughout this paper, provide a normalised measurement for the degree of obesity in children and young people, indicating to what degree an individual BMI lies above or below the median BMI value.

A meta-analysis by Ho *et al*[18] concluded that lifestyle interventions can lead to improvements in weight and cardiometabolic outcomes in child obesity. However, while numerous lifestyle intervention programmes to tackle childhood obesity are conducted across the UK, and many describe statistically significant reductions in BMI-SDS,[19] these results do not necessarily translate into clinical benefit for the individual. How reducing BMI-SDS in a trial translates to a reduction in adiposity is uncertain.

Paediatric weight management guidelines exist in many countries to promote best practice, but at present many of these recommendations are based on low-grade scientific evidence.[20] Understanding how much BMI must be reduced to positively affect body composition and metabolic health is important to ensure that treatment interventions are appropriately designed and evaluated.[21]

Given the scale of the obesity problem and the significant and sustained adverse effects on health, clinically effective paediatric weight management treatment options are vital. A meta-analysis of cardiovascular disease risk in healthy children and its association with BMI has been conducted,[22] but there is yet to be a systematic quantification of the reduction in BMI required to improve adiposity in the paediatric population with obesity.

It is important to highlight that when assessing interventions designed to manage overweight and obesity in children and adolescents, it is essential to recognise that measures such as BMI and derived SDS are surrogates of the real purpose: reduction of adiposity, fat being the key organ involved in metabolic complications.[23] To rigorously assess the clinical and cost-effectiveness of weight management interventions in young people, it is first necessary to understand what BMI-SDS change means in terms of key outcomes such as effects on adiposity. This paper is designed to put BMI-SDS changes in context when considering improvement in adiposity (fatness). Through meta-regression analysis, we explore the potential impact of BMI-SDS reduction in terms of change in percentage body fat. The outcome of which will both inform clinical guidelines for paediatric weight management interventions and guide outcome measures in future clinical trials.

## Objective
This paper aims to establish the minimum change in BMI-SDS needed to effect improvements in adiposity markers of children and adolescents with obesity. This is the first of a series of three papers reporting on the findings from studies identified in a large systematic review (n=90 studies; searched up to May 2017) and focuses on the evidence in relation to adiposity (percentage body fat); the others relating to metabolic and cardiovascular health.

## METHODS
The studies included in this paper were identified as part of large-scale systematic review (PROSPERO CRD42016025317). The protocol for this systematic review is available: https://doi.org/10.1186/s13643-016-0299-0. The final search was conducted in May 2017, the review was completed in January 2018 and the results are still being evaluated.

### Participants
Studies with participants aged 4–19 years with a diagnosis of obesity using defined BMI thresholds were considered for inclusion. BMI-SDS was calculated as a function of the degree of obesity of the subjects when compared with BMI references. BMI standards included, but were not limited to, the 98th percentile on the UK 1990 growth reference chart,[24] 95th percentile on the US Centre for Disease Control and Prevention growth chart,[25] the International Obesity Task Force (IOTF) BMI for age cut-points[26] and the WHO growth references,[27 28] in addition to country-specific obesity thresholds using BMI reference data from their paediatric populations. Studies that included overweight, as opposed to obese, individuals, pregnant females or those with a critical illness, endocrine disorders or syndromic obesity were excluded from this review.

### Interventions
Studies of lifestyle treatment interventions that included dietary, physical activity and/or behavioural components with the objective of reducing obesity were included. Interventions of <2 weeks duration and those that involved surgical and/or pharmacological components (eg, bariatric surgery, drug therapy) were excluded. Studies focused on obesity prevention were also excluded. No restrictions were imposed regarding the setting or delivery of the interventions.

### Outcome measures
To meet the inclusion criteria of the full systematic review, interventions had to report baseline (preintervention) and postintervention BMI-SDS or change measurements of BMI-SDS plus one or more markers of metabolic health (please refer to the published protocol paper for a complete list of the metabolic health markers of interest; https://doi.org/10.1186/s13643-016-0299-0).

This paper focuses on change in BMI-SDS and adiposity measures other than BMI, including waist circumference and percentage body fat.

## Study design
Completed, published, randomised controlled trials (RCTs) and non-randomised studies (cohort studies) of lifestyle treatment interventions for children and adolescents with obesity, with or without follow-up.

## Ethics
Ethical approval was not required as this paper reviewed published studies only.

## Patient and public involvement
There was no patient or public involvement in this review of published studies.

## Information sources and search methods
Studies were identified by searching five electronic databases from inception to May 2017 (AMED, Embase, MEDLINE via OVID, Web of Science and CENTRAL via Cochrane library), alongside scanning reference lists of included articles and through consultation with experts in the field. The search strategy for MEDLINE database is presented in online supplementary appendix 1.

## Study selection and data extraction
Titles and abstracts were assessed for eligibility and the data outcome measures described previously were extracted by two independent reviewers from the review team using a standardised data extraction template, which was piloted by both reviewers before starting the review to ensure consistency.

## Quality assessment
The focus of this study is the relationship between change in BMI-SDS and change in metabolic health parameters, rather than the specific treatment interventions that effect those changes. Therefore, risk of bias tools, such as the Cochrane Risk of Bias tool,[29] were not considered appropriate. The included studies were assessed for methodological quality by two members of the review team during the data extraction process using the Quality Assessment tool used in the 2004 Health Technology Assessment (HTA) systematic review of the long-term effects and economic consequences of treatments for obesity and implications for health improvement.[30] This Quality Assessment tool comprises 20 questions which are added together to give a final score and a percentage rating, from which a level of quality is assigned. Any discrepancies in Quality Assessment scoring were resolved through discussion.

## Analysis
We carried out random-effects meta-regression as implemented in Stata[31] to try to quantify the relationship between mean change in BMI-SDS (independent, predictor variable) and mean change in percentage body fat (target variable), where these were either reported, or

were able to be calculated from reported data. Further details are given below. We were not trying to assess the relative effects of the various interventions, but rather to examine the relationship between these two outcomes. Meta-regression allows for residual heterogeneity in the target variable not explained by the predictor. Subsets from the same study (eg, intervention vs control, boys vs girls, see below) were regarded as independent observations provided there was no data duplication.

## RESULTS
### Search results
In total, 98 published articles relating to 90 different studies met the inclusion criteria for the entire systematic review. See figure 1 for a flow diagram illustrating the number of papers excluded at each stage of the review. For studies reported in multiple publications, the reference that provided the most comprehensive information has been used (see footnote of table 1 for details).

The Venn diagram (figure 2) illustrates how many studies were identified for the various markers of metabolic health. Seventy-three studies assessed and reported adiposity measures. The adiposity measures reported included percentage body fat, body fat-SDS, body mass, fat mass, fat-free mass, waist circumference and waist circumference-SDS. The 68 studies that examined diabetes/inflammation measures (HOMA-IR, insulin, glucose, C reactive protein, interleukin-6, alanine transaminase and the 71 studies examining cardiac measures (eg, lipids, cholesterol, blood pressure) will be reported separately.

### Studies for inclusion in meta-regression analysis
Seventy-three studies assessed and reported adiposity measures. Of the different adiposity measures that were reported in these studies (percentage body fat, body fat-SDS, body mass, fat mass, fat-free mass, waist circumference and waist circumference-SDS), we elected to examine percentage body fat as it was far more frequently reported across studies. Therefore, of the 73 adiposity studies, we conducted our meta-regression on 39 studies which reported percentage body fat values. These studies are presented in table 1 with the corresponding changes in BMI-SDS.

The results of five studies were duplicated in multiple papers, thus the reference that reported the most comprehensive information was used in the analysis; see table 1 footnote for details. Thirty-four studies were excluded from the meta-analysis; the characteristics of the excluded studies, along with the reason for exclusion, are summarised in online supplementary appendix 2.

### Narrative description of studies that reported BMI-SDS and percentage body fat
Of the 39 studies that reported percentage body fat included in our analysis, 7 were conducted in both Germany and the USA, 4 in Italy, followed by Australia (n=2), Denmark (n=2), the Netherlands (n=2), Poland

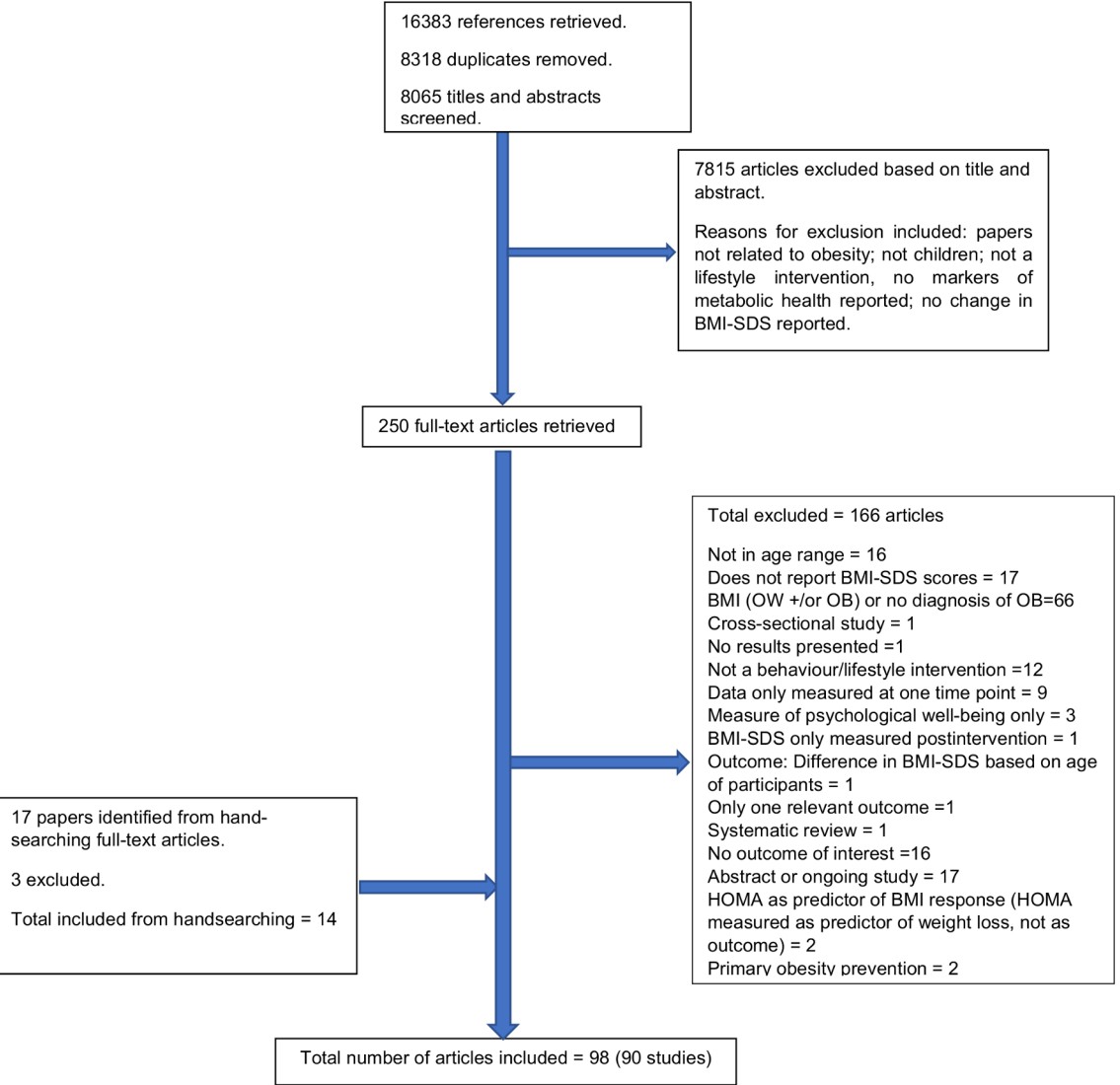

16383 references retrieved.

8318 duplicates removed.

8065 titles and abstracts screened.

7815 articles excluded based on title and abstract.

Reasons for exclusion included: papers not related to obesity; not children; not a lifestyle intervention, no markers of metabolic health reported; no change in BMI-SDS reported.

250 full-text articles retrieved

Total excluded = 166 articles

Not in age range = 16
Does not report BMI-SDS scores = 17
BMI (OW +/or OB) or no diagnosis of OB=66
Cross-sectional study = 1
No results presented =1
Not a behaviour/lifestyle intervention =12
Data only measured at one time point = 9
Measure of psychological well-being only = 3
BMI-SDS only measured postintervention = 1
Outcome: Difference in BMI-SDS based on age of participants = 1
Only one relevant outcome =1
Systematic review = 1
No outcome of interest =16
Abstract or ongoing study = 17
HOMA as predictor of BMI response (HOMA measured as predictor of weight loss, not as outcome) = 2
Primary obesity prevention = 2

17 papers identified from hand-searching full-text articles.

3 excluded.

Total included from handsearching = 14

Total number of articles included = 98 (90 studies)

**Figure 1** Flow diagram from the systematic review that identified the included studies. BMI-SDS: body mass index-SD score; HOMA, homeostatic model assessment (method of assessing insulin resistance); OB: obese; OW: overweight.

(n=2), Switzerland (n=2), Tunisia (n=2) and one each in Belgium, Brazil, Canada, Chile, France, Portugal, Spain, Thailand and the UK. There were country-specific variations in the definition of obesity, with most studies defining obesity by participants having a BMI-SDS >2, or a BMI percentile of at least >90th percentile. Most of studies used a cohort design (n=27), 11 were RCTs, of which 1 included results from a cohort of the original RCT. There was also one study which adopted a quasi-randomised design.

Most studies (n=20) conducted their intervention in the hospital clinic setting. Eight studies conducted the intervention in the community setting and 10 in academic institutions. One conducted the intervention in a mixed setting, reporting use of both a community setting and academic institution.

Twenty-eight studies conducted interventions that comprised both diet and exercise components. The remaining studies (n=11) used interventions that focused either on exercise or diet only. The duration of the interventions ranged from 15 days to 24 months. The majority of studies (n=29; 74%) did not report any follow-up after the lifestyle treatment intervention. The duration of follow-up in the studies where it was conducted and reported, ranged from 6 months to 2 years.

The sample sizes of the included studies ranged from 8 to 203 participants. The age of the participants ranged from 4 to 19 years. Studies predominantly had a mix of males and females (95%) with only three studies specifically focused on either only girls[32 33] or boys.[34] Seventeen studies (44%) measured pubertal development of participants according to Marshall and Tanner staging, with pubertal status categorised into three groups: prepubertal, pubertal and late/postpubertal.[35] Four studies (10%) reported that pubertal development was measured but the methodology was not defined. Eighteen studies (46%) did not report any measures of pubertal development.

**Table 1** Characteristics of studies reporting adiposity outcomes with results of mean change in BMI-SDS and percentage body fat

| | Author, Country, (Intervention name) | Study design: Sample size (n) Analysed (An) | Obesity definition | Age range (Inclusion); Mean ± (SD) Sex (% F) | Pubertal status measured | Diet D/ Exercise (E)/ D+E: Setting | Format & content | Duration (months); Follow up (months) | Method of % body fat measurement | Δ BMI SDS/ z-score by subgroup when reported | Δ % body fat score by subgroup when reported |
|---|---|---|---|---|---|---|---|---|---|---|---|
| 1 | Bell et al[52] Australia | Cohort Total = 14 (14) | BMI ≥95th %ile | Age range: 9–16 12.70(2.32); F=43% | Yes—Tanner | E: community | 8 weeks structured circuits exercise training: 3 x 1hr sessions/week. No standard dietary modifications. | 2: 0 | DXA | All: −0.03 | All: −0.57 |
| 2 | Book et al[53] Canada HIP KIDS | Cohort: Total = 42 (41) | BMI ≥95th %ile (CDC) | Age range: 8–17 12.8 ±3.14; F=50% | Yes—Tanner | D+E: Clinic (Hospital) | Intensive phase (3 months): bi-weekly 90 min counselling. Maintenance phase (9 months): alternating mthly GP or individual sessions (90 mins). Sessions focus on exercise/psychosocial/behavioural aspects. | 12: 0 | BIA | All: −0.04 | All: −1.39 |
| 3 | Bruyndonckz et al[36] Belgium | Quasi-RCT: Total = 61 IG = 33 (27) CG = 28 (21) | BMI ≥97th %ile adolescents <16 years; BMI ≥35 adolescents ≥16 years | Age range: 12–18 IG: 15.4±1.5; F = 79% CG: 15.1±1.2; F=73% | NR | D+E: Clinic (Hospital) | Intervention: Dietary restriction 1500-1800 kcal/day + 2 hrs/day supervised play/lifestyle activities + 2hrs/wk PE + 3 x 40min/wk supervised training session. Control: Usual care. | 10: 0 | Subsample also measured using DXA | IG: −1.21 CG: 0.13 | IG: −11.30 CG: 0.4 |
| 4 | Bustos et al[54] Chile | Cohort: Total = 50 (28 completed) | CDC | Age range: NR 9.5 ±1.9; F=48% | NR | D+E: Academic Institution | Nutrition/behavioural modification session 40 min/wk + PA 50 min x2/wk+ Family support every 15 days for first 2 months, then monthly. | 8: 0 | DXA | All: −0.3 | All: −3.00 |
| 5 | Calcaterra et al[55] Italy | Cohort: Total = 22 (22) | BMI > 95th %ile | Age range: 9–16 13.23 ± 1.76; F=41% | Yes - Tanner | E: Academic Institution | 2 x 90 mins exercise training sessions/ wk | 3: 0 | BIA | All: −0.15 | All: −3.30 |
| 6 | Dobe et al[46] Germany OBELDICKS – mini | Cohort: Total = 103 (103) | >97th to 99.5 percentile | Age range: 4–8 6.1 ±1 F=56% | NR | D+E: Academic Institution | Obeldicks mini: focus on training parents (22.5 hrs for parents, 4.5 hrs for children). Group sessions. Parents+children classes every 4th session, Children's classes: 9 x monthly sessions (30 mins); 1x introduction; 3 x diet; 5 x eating habits Parenting classes: 13 x monthly sessions (1.5 hrs): 1x introduction 1x medicine 3x nutrition 5x eating habits + education tips 3x discussion circle Individual consultation: every 2 months (30 mins) Exercise: 50 x weekly sessions (1.5 hrs) | 12: 0 | BIA | Obeldicks-mini: −0.46 | Obeldicks mini: −3.00 |
| 7 | Farpour-Lambert et al[56] Switzerland | RCT: Total = 44 IG= 22 (22) OC =22 (22) | BMI >97th %ile | Age range: 6–11 8.9 ± 1.5 IG: F=59% OC: F= 68% | Yes | E Clinic (Hospital) | 180 min/wk PA + 135 min/wk PE | 3: 0 | Skinfold measurements | IG: −0.1 CG: 0 | IG: −1.50 CG: 0.80 |
| 8 | Ford et al[48 57] UK | RCT: Total = 106 (91) Gp1 SC = 52 (46) Gp 2 Mandometer = 54 (45) | BMI ≥95th %ile (CDC) | Age range: Mandometer: 9.0–16.9 SC: 9.1–17.5 Mandometer: 12.7±2.2 SC: 12.5±2.3 overall F=56% | Yes | D Clinic (Hospital) | Mandometer device to regulate rate of eating and total intake vs SC | 12: 0 | DXA | IG: −0.36 CG: −0.14 | IG: −4.60 CG: −1.30 |

Continued

**Table 1** Continued

| Author, Country, (Intervention name) | Study design: Sample size (n) Analysed (An) | Obesity definition | Age range (inclusion): Mean ± (SD) Sex (% F) | Pubertal status measured | Diet D/ Exercise (E)/ D+E: Setting | Format & content | Duration (months): Follow up (months) | Method of % body fat measurement | Δ BMI SDS/ z-score by subgroup when reported | Δ % body fat score by subgroup when reported |
|---|---|---|---|---|---|---|---|---|---|---|
| 9 Gajewska et al[57] Poland | Cohort: Total = 100 (76) With WL =71 (56) Without WL = 29 (20) | BMI SDS >2 | Age range: 5–10 with WL: 8.1 (6.8–9.2); F= 51% without WL: 8.8(7.3–9.6); F=59% overall F = 53% | Reported with Tanner stage, any with pubertal development excluded. | D+E; Community & Academic institution | 3-month intervention, low energy diet (1200–1400kcal), 3–5 meals every day, instructions concerning PA, 10–14 food day diary, 3-day food diary. | 3: 0 | BIA | WL: –0.98 No WL:–0.2 | WL: –2.90 No WL:0.30 |
| 10 Garanty-Bogacka et al[58] Poland | Cohort: Total = 50 (50) | BMI >97th %ile (Polish ref pop.) | Age range: 8–18 14.2 ±2.6; F=58% | Yes | D+E; Clinic (Hospital) | Exercise therapy (Instructions in PA + reducing sedentary behaviour) + reduction in fat and sugar intake. | 6: 0 | Skinfold measurements & Lohman's formula | All:–1 | All: –4.70 |
| 11 Gronbaek et al[59] & Kazankov et al[60] Denmark Julemaerkehjemmet Hobro (same cohort) | Cohort: Total = 117 (117) (n=71 attended 12 mth FU) | NR. Obese. BL BMI-SDS: 2.93±0.52 | Age range: NR 12.1 ±1.3 F=56% | NR | D+E; Community | Individually designed healthy diet + moderately strenuous PA program (at least 1hr/day). | 2.5 months/10 weeks: 12 | BIA | All: –0.63 | All: –4.30 |
| 12 Hvidt et al[61] Denmark | Cohort: Total = 61 (61) | Children's Obesity Clinic; BMI >90th %ile (Danish ref pop.) z-score 1.28. BL BMI-SDS: 2.73±0.60 | Age range: 10–18 Median: 12.5 F=54% | NR | D+E; Clinic (Hospital) | Family-centred approach involving behaviour changing techniques (90 advice and advice strategies on low-calorie diet + activity for example, 10–20 items aimed to reduce obesity). | 12: 0 | BIA | All: –0.21 | All: –3.40 |
| 13 Kirk et al[47] USA | Cohort: Total = 177 (177) Children (5–10yrs) = 85 Adolescents (11–19yrs) = 92 | BMI >95th %ile | Age range: 5–19 9.0±1.5 Overall F=61% Children: F = 24% Adolescents: F = 59% | NR | D+E; Clinic (Hospital) | Behavioural intervention with individualised behavioural goals for nutrition, PA & family support. | 5: 6 | DXA | GP1: –0.18 GP2: –0.13 All: –0.15 | GP1: –2.10 GP2: –2.40 All: –2.20 |
| 14 Klijn et al[62] The Netherlands | Cohort: Total = 15 (15) | BMI >30 | Age range: 10–18 14.7 (2.1); F=NR | NR | E; Community | Aerobic exercise training programme – 12 weeks; 3 x 30–60 min aerobic group sessions/week (2x gym/outdoors, 1 x swimming pool). PE teacher led. Diverse indoor, outdoor and swimming activities. | 3: 0 | % body fat calculated by "dividing fat mass by total body mass" | All: –0.4 | All: –3.80 |
| 15 Lazzer et al[63] Italy | Cohort: Total = 19 Boys = 7 (7) Girls = 12 (12) | BMI >97th %ile | Age range: 8–12 Boys: 9.9±1.6 Girls: 11.2±1.5 Overall F=63% | Yes – Tanner | D+E; Community | 2 x 50min/wk endurance training + 2hr/ wk PE lessons + 1 x wk child & parent dietetic class + 1 x wk psychological group class. | 8: 12 | DXA | Boys: –0.4 Girls: –0.2 | Boys: –4.00 Girls: –2.20 |
| 16 Meyer et al[64] Germany | RCT: Total = 67 IG=33 (33) OC=34 (34) | BMI >97th %ile (German paediatric population) | Age range: 11–16 IG: 13.7±2.1; F=48% OC: 14.1±2.4; F =50% | Yes - Tanner | E; Clinic (Hospital) | 3 x exercise sessions (Monday: swimming and aqua aerobic training 60 min + Wednesday sports games 90 min + Friday walking 60 min)/ wk; Control: Maintain current level of PA | 6: 0 | BIA | IG: –0.43 CG: –0.14 | IG: –1.00 CG: 0.00 |
| 17 Miraglia et al[65] Brazil | Cohort: Total = 27 (27) | BMI z-score >2 | Age range: 6–13 Median 10.3; F=48% | NR | D+E; Clinic (Hospital) | AmO: Outpatient Ambulatory. Obesity outpatient clinic - lifestyle change based on goals agreed relative to feeding habits & physical exercise, followed mthly. 12 months: Subjects assessed at inclusion & after 12 months of FU to obtain anthropometric & adipokine measurements. | 12: 0 | BIA | All: –0.4 | All: –0.10 |

Continued

**Table 1** Continued

| Author, Country, (Intervention name) | Study design: Sample size (n) Analysed (An) | Obesity definition | Age range (inclusion): Mean ± (SD) Sex (% F) | Pubertal status measured | Diet D/ Exercise (E/ D+E: Setting | Format & content | Duration (months): Follow up (months) | Method of % body fat measurement | Δ BMI SDS/ z-score by subgroup when reported | Δ % body fat score by subgroup when reported |
|---|---|---|---|---|---|---|---|---|---|---|
| 18 Morell-Azanza et al[66] & Rendo-Urteaga et al[67] Spain (same cohort) | Cohort: Total = 54 (40) high responders =21 low responders = 19 | OW/OB as per Cole et al 2000 | Age range: 7–15 Mean = 11 F=53% (of N analysed) | Yes – Tanner | D: Clinic (Hospital) | Moderate energy-restricted diet + nutritional education sessions with dietitian + family involvement. | 2.5; 0 | BIA | HR: −0.79 LR: −0.18 HR: −0.64 LR: −0.07 | HR: −3.10 LR: −0.60 HR: −2.49 LR: −0.37 |
| 19 Murer et al[68] & Aeberli et al[69] Switzerland (same cohort) | Cohort: Total = 206 (203) | BMI >98th %ile | Age range: 10–18 14.1±1.9; F=44% | NR | D+E: Clinic, hospital | Moderate caloric restriction.2 x 60–90 min/day endurance exercise + 4–5 hr/wk. exercise session + behaviour modification. | 2; 0 | BIA | All: −0.42 | All: −5.50 |
| 20 Murdolo et al[70] Italy | Cohort: Total = 53(53) Responders = 44 Non-responders = 9 | NR | Age range: 5–13 Responders: 9.0±1.1; F=50% Non-responders: 2.09±0.32; F=33% | Yes – Tanner | D+E: Community | Educational Wt Excess Reduction Program | 24; >6 | BIA | Responders: −0.44 Non-responders: 0.11 | Responders:−2.90 Non-responders: −2.00 |
| 21 Ning et al[71] & BEAN et al[72] USA TEENS (same cohort) | Cohort: Total = 145**(145) | BMI ≥95th %ile (CDC) | Age range: 11–18 13.1 F=65% | NR | D+E: Academic Institution | 12 x 30 min nutritional session with adolescent and parent/s + Education/ behavioural support sessions once every 2 wks, or alternating wks + PA 3 x 60 min/wk during initial 12 wks, then minimum of twice/wk. | 6; 0 | DXA | All: −0.1 | All: −2.40 |
| 22 Pacifico et al[73] Italy | Cohort: Total = 120 (120) | BMI >95th %ile | Age range: (11.5–12.2) 11.9; F=35% | Yes (method ND) | D+E: Clinic (Hospital) | Hypocaloric diet (25–30 Kcal/kg/day) + 60 min/day ~ 5 days/wk moderate exercise + Reduce sedentary behaviour. | 12; 0 | NR | All: −0.32 | All: −2.10 |
| 23 Racil et al[32] Tunisia | RCT: Total = 34 HIIT = 11 (11) MIIT = 11 (11) OC = 12 (12) | BMI >97th %ile (French standards) | Age range: NR HIIT: 15.6±0.7 MIIT: 16.3±0.52 OC:15.9±1.2 Overall F=100% | Yes –Tanner | D+E: Community | 4-day diet records + HIIT or MIIT. Interval training program 3 x /wk on non-consecutive days. | 3; 0 | BIA | HIT: −0.4 MIT: −0.3 OC: 0 | HIT: −2.90 MIT: −2.00 OC: −0.40 |
| 24 Racil et al[33] Tunisia | RCT: Total = 47 HIIT = 17 (17) MIIT = 16 (16) OC = 14 | BMI >97th %ile (French standards) | Age range: NR 14.2±1.2; F=100% | NR | E: Academic Institution | HIIT (Warm up + Interval training at 100%/50% MAS + Cooling down); MIIT (Warm up + Interval training 80%/50% MAS + Cooling down) | 3; 0 | BIA | HIT: −0.3 MIT: −0.3 OC: 0 | HIT:−3.90 MIT: −3.40 OC: −0.50 |
| 25 Reinehr et al[38] Germany OBELDICKS | Cohort: Total = 42 (42) | BMI ≥97th %ile | Age range: 6.1–15.1 10.2; F=57% | Yes - Tanner | D+E: Clinic (Hospital) | Obeldicks: Intensive phase 3 months (Parents' course 2x/month + Behaviour therapy 2x/month + Nutritional course 2x/month + Exercise therapy 1x/wk) + Establishing phase 3 months (Talk rounds for parents 1x/month + Psychological therapy + Exercise therapy 1x/wk) + Establishing phase 2 for 3 months (Psychological therapy + Exercise therapy 1x/wk) + Establishing phase 3 for 3 months (Exercise therapy 1x/wk). | 12; 0 | % body fat skinfold thickness | Sig. WL −0.9 NS WL: −0.2 | Sig. WL:−7.50 NS WL: −3.00 |
| 26 Reinehr et al[74][75] Germany OBELDICKS | Cohort: Ob + Sub. WL = 25 Ob + no change = 18 Normal control = 19 (BL data only) | IOTF using pop.-specific data | Age range: Ob: 10.8±2.6; F=61% Lean C: 10.3±2.9; F=58% Ob + Sub. WL : F= 68% Ob + no change: F = 50% | Yes -Tanner | D+E: Clinic (Hospital) | Obeldicks | 12; 0 | % body fat skinfold thickness | WL: −0.6 No WL: −0.1 | WL: −8.00 No WL: 0.00 |

Continued

**Table 1** Continued

| Author, Country, (Intervention name) | Study design: Sample size (n) Analysed (An) | Obesity definition | Age range (inclusion); Mean ± (SD) Sex (% F) | Pubertal status measured | Diet D/ Exercise (E)/ D+E: Setting | Format & content | Duration (months); Follow up (months) | Method of % body fat measurement | Δ BMI SDS/ z-score by subgroup when reported | Δ % body fat score by subgroup when reported |
|---|---|---|---|---|---|---|---|---|---|---|
| 27 Rohner et al[76] Germany **Fit Kids** | Cohort: Total = 22 (22) Unchanged BMI= 12 Reduced BMI = 10 | BMI >99.5th %ile (German standard values) or BMI >97th %ile with obesity-associated risk factors or BMI >90th %ile with obesity-associated disease | Age range: 7-15 Median: 11.9 F=27% Unchanged BMI: F =33% Reduced BMI: F=20% | NR | D+E: Community | Physical exercise (2 x wk, 100 hrs in total) + Nutritional/heath education and psychological care for the child (x wk, 43.5 hrs total) and parent/s (2 x wk, 12 hrs total). | 12: 0 | BIA | Increased BMI: 0.12 Reduced BMI: -0.35 | Increased BMI: 1.05 Reduced BMI:-0.05 |
| 28 Rolland-Cachera et al[77] France | RCT: Total = 99 PROT- = 61 (53) PROT+ =60 (46) | BMI > 97th %ile (French reference values) | Age range: 11-16 PROT-= 14.1±1.2; F = 74% PROT ± =14.4±1.3; F = 72% | NR | D+E: Academic Institution | Wt reducing diet; 7hr/wk vigorous sports + 7hr/wk outdoor activities; advice on nutrition & PA during wkends/ holidays. | 9: 12+24 | BIA | PROT-:-2.6 PROT+:-2.5 | PROT-:-12.40 PROT+:-12.10 |
| 29 Roth et al[78] Germany **OBELDICKS** | Cohort: Total = 69 OB + WL = 32 OB + with WL = 37 | OB as per IOTF criteria | NR – (see Obeldicks age range) Ob with WL: 11.8±2.0; F=50% Ob without WL: 12.1±2.1; F=51% Normal wt: 12.3±3.0; F=45% | Yes - Tanner | D+E: Clinic (Hospital) | Obeldicks | 12: 0 | % body fat skinfold thickness | WL: -0.69 No WL: 0.03 | WL: -9.60 No WL: -4.30 |
| 30 Savoye et al[79] USA **Bright Bodies** | Cohort: Total = 33 (25) SMP = 10 (8) BFC = 23 (17) | BMI ≥95th %ile | Age range: 11-16 13.5±0.3; SMP:13.3±0.6; F=75% BFC: 13.6±0.3; F= 65% | NR | D+E: Academic Institution | Bright Bodies Weight Management Program: nutrition education, exercise, behavioural modification. 2 x 30 min exercise sessions + 1 x 45 min nutrition/behaviour group session per week. 4 levels: Beginner, Intermediate i, Intermediate ii, Advanced. All levels 12 weeks duration. Monthly maintenance classes after 1 yr (support-group style) | 12: 12 | BIA | SMP: -0.36 BFC: -0.12 | SMP:-6.50 BFC: -4.20 |
| 31 Savoye et al[80 81] USA **Bright Bodies** (data taken from 2011 paper) | RCT+ Long term FU results (cohort) RCT Total = 174 BB = 105 CC = 69 1 YR ANALYSIS BB = 75 CC = 44 | BMI ≥95th %ile (CDC) | Age range: 8-16 BB: 12.0±2.5; F=56% CC: 12.5±2.3; F=68% | NR | D+E: Academic Institution (local school). | Bright Bodies Weight Management Program: nutrition education, exercise, behavioural modification. 2 x sessions/ wk for 6 months, then biweekly for next 6 months. BB: 2x50 min exercise + 1x40 min nutrition/behaviour modification per wk + 12 months no active intervention. Control group: standard care – paed. obesity clinic (biannual clinic appt; diet + exercise counselling) Structured tx & teaching program (28 x 45 min therapeutic sessions for example, PA, nutrition, healthy cooking) | 12: 12 FU 1.5: 24 | BIA | IG: -0.21 CG: 0.01 | IG: -3.90 CG: 2.10 |
| 32 Savoye et al[82] USA **Bright Bodies** | RCT Total = 75 BB = 38 (31) CC = 37 (27) | BMI ≥95th %ile | Age range: 10-16 BB: 12.7 (1.9); F=68% CC: 13.2 (1.8); F=62% | Yes-Tanner | D+E: Academic Institution | Bright Bodies Weight Management Program: nutrition education, exercise, behavioural modification. 2 x 30 min exercise sessions + 1 x 45 min nutrition/ behaviour medication group session per week. 4 levels: Beginner, Intermediate I, Intermediate ii, Advanced. All levels 12 weeks duration. Monthly maintenance classes after 1 yr (support-group style) | 6: 0 | BIA | BB: -0.05 CC: 0.04 | BB: -3.30 CC: 0.40 |

Continued

**Table 1** Continued

| | Author, Country, (Intervention name) | Study design: Sample size (n): Analysed (An) | Obesity definition | Age range (inclusion): Mean ± (SD): Sex (% F) | Pubertal status measured | Diet D/ Exercise (E)/ D+E: Setting | Format & content | Duration (months): Follow up (months) | Method of % body fat measurement | Δ BMI SDS/ z-score by subgroup when reported | Δ % body fat score by subgroup when reported |
|---|---|---|---|---|---|---|---|---|---|---|---|
| 33 | Schiel et al[83] Germany | Cohort: Total = 143 (143) | BMI-SDS ≥97th %ile | Age range: NR 13.9±2.4; F=62% | NR | D+E; Clinic (Hospital) | Structured Tx & Teaching Program (STTP): 28 x 45 min therapeutic sessions for example, PA, nutrition, healthy cooking | 1.5: 24 | NR | All: -0.26 | All: -3.40 |
| 34 | Seabra et al[84] Portugal | Cohort: Total = 88 soccer = 29 (29) Trad. act. = 29 (29) OC = 30 (30) | BMI-SDS > 2 | Age range: 8-12 Soccer: 10.5±1.5; Trad. act: 11.0±1.6; OC=10.0±1.3 Overall F=0% | Yes - Tanner | E; Community | Soccer & trad. activity programmes (3 x 60-90min/wk) + 2 x 1hr at BL & 3 months later energy balance session. | 6: 0 | DXA | Soccer: -0.2; Trad.: -0.2; CG: -0.1 | Soccer:-2.20; Trad:-4.10; CG:3.10 |
| 35 | Truby et al[84] Australia | RCT: Total = 87 SMC = 37 (33) SLF = 36 (32) WList OC = 14 (14) | BMI >90th %ile (CDC) | Age range: 10-17 SMC: 13.2±1.9; F=73%; SLF: 13.2±2.1; F=72%; WList OC: 13.6±1.9; F=71% | Yes -Tanner | D; Clinic (Hospital) | Structured modified CHO diet (35% CHO; 30% protein; 35% fat), structured low-fat diet (55% CHO; 20% protein; 25% fat). Control (no dietary advice). | 3: 0 | BIA | SLF: -0.09; SMC:-0.15; CG: 0.02 | SLF: -0.13; SMC: -0.40; CG: 2.62 |
| 36 | Van der Baan-Slootweg et al[85] The Netherlands | RCT: Total = 90 Inpt. = 45 (37) AmO = 45 (36) | BMI z score ≥ 3.0 or > 2.3 with OB-related health problems | Age range: 8-18 Inpt: 13.8±2.3; F=58%; AmO: 13.9±2.5; F=58% | NR | D+E; Clinic (Hospital) | Inpt. (Hospitalised 26 wks on working days - 4 days/wk 30-60min exercise + nutrition/BM once/wk + parents/caregivers 3 x 1hr lesson on nutrition/BM); Ambulatory (12 visits at increasing time intervals - 1 hr exercise session + encouraged 3 x exercise/wk + 1 hr educational programme + 30 min nutrition education). | 6: 24 | BIA | InpT: -0.6; AmO: -0.35 | InP: -3.34; AmO:-7.87 |
| 37 | Visuthranukul et al[86] Thailand | RCT: Total = 70 (52) I = 35(25) OC = 35 (27) | ND. BL BMI z-score: I = 3.7±0.9; C = 3.6±1.6 | Age range: 9-16 I = 11.9±1.9; F=36% C = 12.0±2.1; F=30% | Yes -Tanner | D; Clinic (Hospital) | I (Low GI diet + Energy restriction 1400-1500 kcal/day + Increased exercise); OC (Energy restriction 1200-1300 kcal/day + Low fat/high fibre diet + Increased exercise). | 6: 0 | BIA | IG:-0.3; CG: -0.3 | IG:0.10; CG:0.10 |
| 38 | Vitola et al[87] USA | Cohort: Total = 8(7) | BMI ≥95th %ile | Age range: NR 15.3±0.6; F=12.8% | Yes -Tanner | D+E; Clinic (Hospital) | Individual behavioural therapy sessions with psychologist. Parents involvement encouraged. Self-monitoring of PA & food intake. Gradual reduction of caloric intake to ≈1200-1500 kcal/day. Ongoing therapy - wt loss therapy repeated when 5% body wt lost & wt stable for at least 4 wks | NR | DXA | All: -0.3 | All: -5.30 |
| 39 | Wickham et al[88] & Evans et al[89] USA TEENS (same cohort) | Cohort: Total = 168 (64)** Completers only = 57 | BMI ≥95th %ile (CDC) | Age range: 11-18 13.9±1.9; F=62% | NR | D+E; Academic Institution | Exercise 1 day/wk at facility + 2 additional exercise days at facility of ppts' choice + 30 min/wk nutrition education/behavioural support sessions. | 6: 0 | BIA | Completers: -0.07 | Completers:-1.30 |

For studies reported in multiple publications, the reference that provided the most comprehensive information has been used (thus Ning et al[71] includes data from Bean et al[72], Evans et al[89] is reported under Wickham et al[88], Aeberli et al[69] is reported under Murer et al[68], Rendo-Urteaga et al[67] is reported under Morell-Azanza et al[66] and Kazankov et al[80] is reported under Grønbæk et al[59].

*studies with change in % body fat included in the analysis.
**Minor discrepancies in reporting of data in papers.

**KEY:** %ile, percentile; AmO, outpatient ambulatory; An., analysed; apt., appointment; BB, Bright Bodies; BIA, bioelectrical impedance analysis; BFC, better food choices; BL, baseline; BM, behaviour modification; BMI, body mass index; C, control; CBT, cognitive behavioural therapy; CDC, Centre for Disease Control; CG, control group; CHO, carbohydrate; D, diet; DXA, Dual-energy X-ray absorption; E, exercise; FBBT, family-based behavioural treatment; F, female; FU, follow up; GI, glycaemic index; GT, group therapy; HGI, high glycaemic index; HIIT, high intensity interval training; hr, hour; HZ, heterozygous; HO, homozygous; ht, height; I, intervention group; IG, intervention group; IOTF, International Obesity Task Force; Inpt., inpatient; LGI, low glycaemic index; LMS, least-mean-squares; LS, long stay; MAS, maximal aerobic speed; MIIT, moderate intensity interval training; min, minute; MO, moderately obese; norm., normal; n, number; NAFLD, Non-alcoholic fatty liver disease; ND, not described; NR, not reported; OB, obese; OC, obese control; OW, overweight; paed, paediatric; PA, physical activity; PROT, protein; RCT, randomised controlled trial; SD, standard deviation; SDS, standard deviation score; SMP, structured meal plan; SS, short stay; Sub., substantial; SMC, structured modified carbohydrate diet; trad., traditional; trad. act, traditional activity; tx, treatment; wk, week; WList OC, wait list obese control; WL, weight loss; wt, weight; X-over, crossover; yr, year.

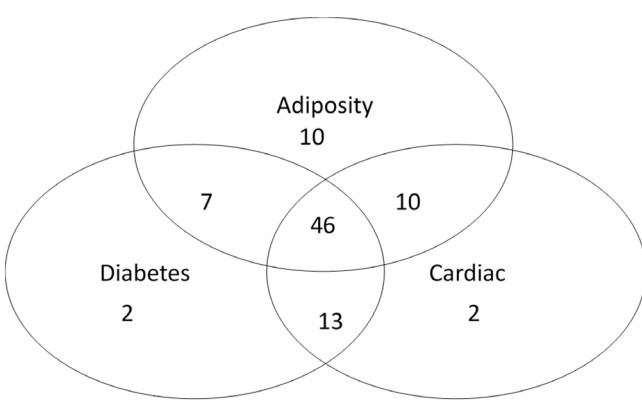

Adiposity
10

7        46        10

Diabetes
2

Cardiac
2

**TOTAL NUMBER OF STUDIES = 90**

**Figure 2**  Venn diagram illustrating the markers of metabolic health measured.

## Quality assessment

The quality of the conduct of each study was assessed using the same criteria as the HTA systematic review of the long-term effects and economic consequences of treatments for obesity and implications for health improvement.[30] The results of the quality assessment can be found in table 2. In summary, none of the 39 studies that reported percentage body fat were considered to be of poor quality, 21 studies (54%) were rated as being of moderate quality and 18 studies (46%) achieved a score over 81% indicating high quality.

## Quantitative analysis

From the 39 studies we identified all data subsets that reported a mean change in BMI-SDS, an associated mean change in percentage body fat (or prestudy and post-study values from which these could be calculated) and the number of cases analysed. A few studies yielded only aggregated data for the whole study. For the others, typical data subsets included intervention versus control, male versus female or good versus poor responders (table 1), and these were used in preference to aggregated results if both were available. In all, there were 66 subsets, with numbers analysed totalling 2618.

SEs were required for the mean changes in percentage body fat and, if not given explicitly, were calculated, from either the SDs or the 95% CIs of the mean changes. In total, 22 data sets had SEs. For the remainder, the SEs were estimated from the SDs associated with the baseline and the postintervention percentage body fat values, making an assumption about the degree of correlation between them. The median and IQR of the correlation coefficients estimated from the nine data sets where both the SEs of mean change and the SDs for baseline and postintervention percentage body fat values were available was 0.81 (IQR 0.59–0.82) and 0.81 has been used in the following analysis.

A small number of data sets (n=6)[36–38] only had medians and IQRs (or range) reported for the baseline

and postintervention results; the mean and SDs were estimated from them.[39]

The meta-regression line was fitted and plotted together with the 95% prediction intervals for the change in percentage body fat across the study data sets. The smallest reduction of mean BMI-SDS associated with a reduction in mean percentage body fat was determined as the smallest reduction in mean BMI-SDS with an associated 95% prediction interval wholly below zero.

A series of sensitivity analyses were conducted. Sensitivity analysis 5A: using the 22 cases where the SEs of the mean change in percentage body fat were actually known, sensitivity analysis 5B: omission of two extreme values and sensitivity analysis 5C: assuming a correlation of 0.50 instead of 0.81. In further exploratory analyses, the percentage of girls and the length of the study (baseline to end of intervention) were added to see if these affected the prediction of mean change in percentage body fat.

## Results from the quantitative analysis

Figure 3 shows the results of the analysis and the fitted regression line. The circles represent the study results (ie, the mean changes in percentage body fat and mean changes in BMI-SDS) analysed for each study, with the size of the circles representing the precision of the mean change in percentage body fat, that is, the reciprocal of the SE squared.

The fitted regression line shown in figure 3 is:

Mean change in percentage body fat=5.179×mean change in BMI-SDS−0.767.

The regression slope was statistically significant (p<0.001), confirming a relationship between the mean loss of percentage body fat and the mean change in BMI-SDS across the data subsets; the proportion of the between-subset variance explained by the mean change in BMI-SDS (ie, 'a type of adjusted R-squared') was 68%. There was, however, significant between-subset heterogeneity with 89% of the percentage of the total residual variance attributable to this (ie, $I^2$).[2] It was further noted that when added to the model, neither the percentage girls in the study sets nor the durations of the interventions significantly improved the prediction of mean change in percentage body fat from the mean change in BMI-SDS (p=0.36, p=0.89, respectively).

Figure 3 also shows the 95% prediction intervals for the mean change in percentage body fat. The upper limit of the prediction interval was below 0 only when the mean reduction in BMI-SDS was >0.6, suggesting that any new study should aim to reduce the BMI-SDS by at least this amount to be confident of achieving a mean reduction of percentage body fat.

A normal plot for the standardised predicted random effects is shown in figure 4. Most were within ±2, although the data sets themselves were not wholly independent (as some came from the same studies).

None of the sensitivity analyses conducted (figure 5) significantly altered the findings, namely that a mean change of 0.6 or more in BMI-SDS was associated with a

**Table 2** Quality assessment of included studies

| Study | Sample | | | | | | Conduct of study | | | | Follow-up | | | Analysis | | | | Interpretation | | | Total (x/40) | Overall rating |
|---|---|---|---|---|---|---|---|---|---|---|---|---|---|---|---|---|---|---|---|---|---|---|
| | 1. Aims clearly stated | 2. Sample size justified | 3. Age of participant defined | 4. Measurements at start clearly stated? | 5. Measurements likely to be valid and reliable? | 6. Risk factors recorded clearly? | 7. Was the intervention before follow-up defined? | 8. Setting of the study clear? | 9. Is mode of assessment described? | 10. Did untoward events occur during the study? | 11. Was there a follow-up? | 12. Was follow-up necessary? | 13. Are losses to follow-up defined? | 14. Was basic data adequately described? | 15. Do numbers add up? | 16. Did analysis allow for passage of time? | 17. Was statistical significance assessed? | 18. Were the main findings interpreted adequately? | 19. Were null/negative findings interpreted? | 20. Are important effects overlooked? | | |
| 1 Bell et al[52] | Yes | Yes | Yes | Yes | Yes | No | Yes | ? | Yes | No | No | Yes | Yes | Yes | Yes | Yes | Yes | Yes | Yes | No | 35 | 87.5 |
| 2 Bock et al[53] | Yes | No | Yes | Yes | Yes | No | Yes | ? | Yes | No | No | Yes | Yes | Yes | Yes | Yes | Yes | Yes | Yes | No | 36 | 90 |
| 3 Bruyndonckx et al[36] | Yes | No | Yes | Yes | Yes | No | Yes | Yes | Yes | No | Yes | Yes | Yes | Yes | Yes | Yes | Yes | Yes | Yes | No | 36 | 90 |
| 4 Bustos et al[54] | Yes | No | Yes | Yes | Yes | No | Yes | Yes | Yes | No | No | Yes | No | Yes | Yes | Yes | Yes | Yes | No | No | 30 | 75 |
| 5 Calcaterra et al[55] | Yes | No | Yes | Yes | Yes | No | Yes | Yes | Yes | No | No | Yes | No | Yes | Yes | Yes | Yes | Yes | ? | ? | 31 | 77.5 |
| 6 Dobe et al[46] | ? | No | Yes | ? | ? | No | Yes | ? | ? | No | No | ? | No | Yes | ? | Yes | Yes | Yes | ? | No | 26 | 65 |
| 7 Farpour-Lambert, et al[56] | Yes | Yes | Yes | Yes | Yes | No | Yes | ? | Yes | No | Yes | Yes | Yes | Yes | Yes | Yes | Yes | Yes | Yes | No | 37 | 92.5 |
| 8 Ford et al[48 57] | Yes | Yes | Yes | Yes | Yes | No | Yes | Yes | Yes | ? | Yes | Yes | Yes | Yes | Yes | Yes | Yes | Yes | Yes | ? | 35 | 87.5 |
| 9 Gajewska et al[37] | Yes | No | Yes | Yes | Yes | No | Yes | Yes | Yes | No | No | Yes | No | Yes | Yes | Yes | Yes | Yes | ? | No | 31 | 77.5 |
| 10 Garanty-Bogacka et al[58] | Yes | No | Yes | Yes | Yes | ? | ? | ? | Yes | ? | No | Yes | No | Yes | ? | Yes | Yes | Yes | ? | No | 26 | 65 |
| 11 Gronbaek et al[59] Kazankov et al[60] | Yes | ? | Yes | Yes | Yes | No | Yes | Yes | Yes | No | Yes | Yes | Yes | Yes | Yes | Yes | Yes | Yes | Yes | No | 37 | 92.5 |
| 12 Hvidt et al[61] | Yes | No | Yes | Yes | Yes | No | Yes | Yes | Yes | No | Yes | Yes | Yes | Yes | Yes | Yes | Yes | Yes | Yes | ? | 34 | 85 |
| 13 Kirk et al[47] | Yes | No | Yes | Yes | Yes | No | Yes | Yes | Yes | ? | No | Yes | ? | Yes | Yes | Yes | Yes | Yes | ? | ? | 29 | 72.5 |
| 14 Klijn et al[62] | Yes | No | Yes | Yes | Yes | No | ? | ? | Yes | ? | No | ? | No | No | Yes | Yes | Yes | Yes | No | No | 27 | 67.5 |
| 15 Lazzer et al[63] | Yes | No | Yes | Yes | Yes | No | Yes | Yes | Yes | No | No | Yes | No | Yes | Yes | Yes | Yes | Yes | Yes | No | 32 | 80 |
| 16 Meyer et al[64] | Yes | No | Yes | Yes | Yes | No | No | Yes | Yes | No | Yes | Yes | No | Yes | Yes | Yes | Yes | Yes | Yes | Yes | 30 | 75 |
| 17 Miraglia et al[65] | Yes | No | No | Yes | Yes | No | Yes | ? | Yes | No | No | ? | No | ? | Yes | Yes | Yes | ? | Yes | No | 25 | 62.5 |
| 18 Morell-Azanza et al[66] Rendo-Urteaga et al[67] | Yes | Yes | Yes | Yes | Yes | No | Yes | Yes | Yes | No | Yes | Yes | No | Yes | Yes | Yes | Yes | Yes | No | No | 32 | 80 |
| 19 Murer et al[68] Aeberli et al[69] | Yes | Yes | Yes | Yes | Yes | No | Yes | Yes | Yes | No | Yes | Yes | Yes | Yes | Yes | Yes | Yes | Yes | Yes | No | 38 | 92 |
| 20 Murdolo et al[70] | Yes | No | Yes | Yes | Yes | No | No | No | Yes | No | Yes | Yes | No | Yes | Yes | Yes | Yes | Yes | No | No | 28 | 70 |
| 21 Ning et al[71] Bean et al[72] | Yes | No | Yes | Yes | Yes | No | Yes | Yes | Yes | No | Yes | Yes | Yes | Yes | Yes | Yes | Yes | Yes | No | No | 34 | 85 |
| 22 Pacifico et al[73] | Yes | No | Yes | Yes | Yes | No | Yes | Yes | Yes | No | Yes | Yes | No | Yes | Yes | Yes | Yes | Yes | No | ? | 31 | 77.5 |
| 23 Racil et al[82] | Yes | No | Yes | Yes | Yes | No | Yes | ? | Yes | No | No | ? | ? | Yes | Yes | Yes | Yes | Yes | Yes | ? | 29 | 72.5 |
| 24 Racil et al[83] | Yes | No | ? | Yes | Yes | No | Yes | ? | Yes | No | No | Yes | No | Yes | Yes | Yes | Yes | Yes | No | No | 28 | 70 |
| 25 Reinehr et al[38] | Yes | No | Yes | Yes | Yes | No | Yes | ? | Yes | No | No | Yes | No | Yes | Yes | Yes | Yes | Yes | No | ? | 29 | 72.5 |
| 26 Reinehr et al[74 75] | Yes | No | Yes | Yes | Yes | No | Yes | Yes | Yes | No | No | Yes | No | Yes | Yes | Yes | Yes | Yes | Yes | No | 32 | 80 |
| 27 Rohrer et al[76] | Yes | No | Yes | Yes | Yes | No | Yes | Yes | Yes | No | Yes | Yes | No | Yes | Yes | Yes | Yes | Yes | ? | No | 33 | 82.5 |
| 28 Rolland-Cachera et al[77] | Yes | No | Yes | Yes | Yes | No | Yes | Yes | Yes | No | Yes | Yes | Yes | ? | Yes | Yes | Yes | Yes | No | No | 33 | 82.5 |
| 29 Roth et al[78] | Yes | No | Yes | Yes | Yes | No | Yes | No | Yes | No | No | Yes | No | Yes | Yes | Yes | Yes | Yes | No | No | 28 | 70 |

Continued

**Table 2** Continued

| Study | Sample | | | | | | Conduct of study | | | | Follow-up | | | Analysis | | | | Interpretation | | | Total (x/40) | Overall rating |
|---|---|---|---|---|---|---|---|---|---|---|---|---|---|---|---|---|---|---|---|---|---|---|
| | 1. Aims clearly stared | 2. Sample size justified | 3. Age of participant defined | 4. Measurements at start clearly stated? | 5. Measurements likely to be valid and reliable? | 6. Risk factors recorded clearly? | 7. Was the intervention before follow-up defined? | 8. Setting of the study clear? | 9. Is mode of assessment described? | 10. Did untoward events occur during the study? | 11. Was there a follow-up? | 12. Was follow-up necessary? | 13. Are losses to follow-up defined? | 14. Was basic data adequately described? | 15. Do numbers add up? | 16. Did analysis allow for passage of time? | 17. Was statistical significance assessed? | 18. Were the main findings interpreted adequately? | 19. Were null/ negative findings interpreted? | 20. Are important effects overlooked? | | |
| 30 | Savoye et al[79] Yes | No | Yes | Yes | Yes | No | Yes | Yes | Yes | No | Yes | Yes | Yes | Yes | Yes | Yes | Yes | Yes | No | No | 34 | 85 |
| 31 | Savoye et al[80,81] Yes | Yes | Yes | Yes | Yes | No | Yes | Yes | Yes | No | Yes | Yes | Yes | Yes | Yes | Yes | Yes | Yes | No | No | 36 | 90 |
| 32 | Savoye et al[82] Yes | Yes | Yes | Yes | Yes | No | Yes | ? | Yes | No | No | Yes | Yes | Yes | Yes | Yes | Yes | Yes | Yes | No | 35 | 87.5 |
| 33 | Schiel et al[83] Yes | No | ? | Yes | Yes | No | Yes | Yes | Yes | No | Yes | Yes | Yes | Yes | Yes | ? | Yes | No | ? | Yes | 29 | 72.5 |
| 34 | Seabra et al[84] Yes | Yes | Yes | Yes | Yes | No | Yes | Yes | Yes | No | No | Yes | Yes | Yes | Yes | Yes | Yes | Yes | No | No | 34 | 85 |
| 35 | Truby et al[84] Yes | Yes | Yes | Yes | Yes | No | Yes | Yes | Yes | No | Yes | Yes | Yes | Yes | Yes | Yes | Yes | Yes | Yes | No | 38 | 95 |
| 36 | van der Baan-Slootweg et al[85] Yes | Yes | Yes | Yes | Yes | No | Yes | Yes | Yes | No | Yes | Yes | Yes | Yes | Yes | Yes | Yes | Yes | No | No | 36 | 90 |
| 37 | Visuthranukul et al[86] Yes | Yes | Yes | Yes | Yes | No | Yes | Yes | Yes | No | Yes | Yes | Yes | Yes | Yes | Yes | Yes | Yes | Yes | No | 38 | 95 |
| 38 | Vitola et al[87] Yes | No | Yes | Yes | Yes | No | ? | ? | Yes | No | No | Yes | No | Yes | Yes | Yes | Yes | Yes | No | No | 28 | 70 |
| 39 | Wickham et al[88] Evans et al[89] Yes | No | Yes | Yes | Yes | No | Yes | Yes | Yes | No | Yes | Yes | ? | Yes | Yes | Yes | Yes | No | Yes | ? | 30 | 75 |

For Q6. Were risk factors clearly recorded? We said 'no' rather than 'unclear' to all the studies that did not record risk factors.
For Q10. Did untoward events occur during the study? We said 'no' rather than unclear if not mentioned.
Rating: not satisfactory 1%–50%; moderate quality=51%–80%; high quality=81%.
?, unclear.

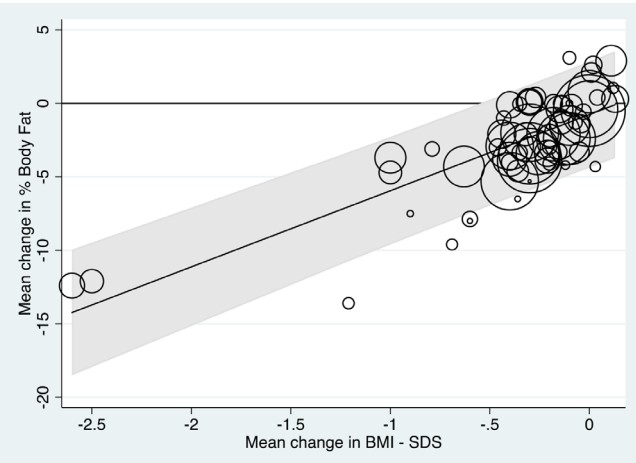

**Figure 3** Meta-regression line showing the relationship between mean change in percentage body fat and body mass index-SD score (BMI-SDS) across the 39 studies (66 subsets) analysed.

definitive mean loss in percentage body fat. In figure 5B, with the exclusion of the two extreme data points, the linear trend can be seen more clearly across the range of mean BMI-SDS losses.

## DISCUSSION
### Summary of main results
This is the first of a series of papers that report on studies identified in a large systematic review. The objective of this paper was to attempt to establish the minimum change in BMI-SDS needed to achieve improvements in body fat in children and adolescents with obesity; BMI-SDS being by far the most frequently reported outcome in terms of weight management trial interventions in childhood. Seventy-three of the 90 included studies reported adiposity measures, but in our meta-regression only percentage body fat can be used as a reliable, comparable marker of change of adiposity. Thus, the analyses presented in this paper were conducted using data from 39 studies. All of

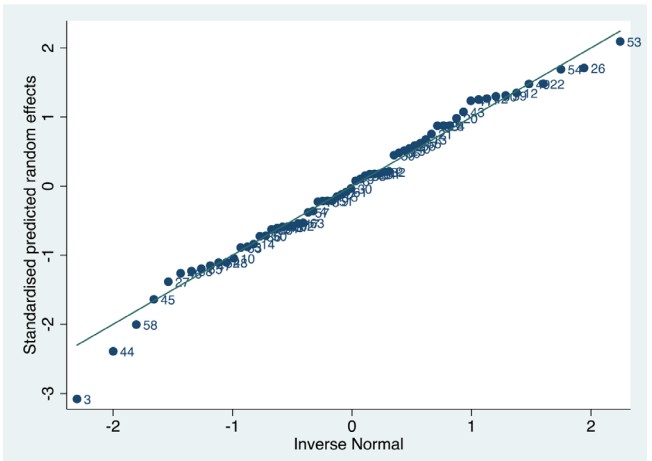

**Figure 4** Normal plot for the standardised predicted random effects from the meta-regression.

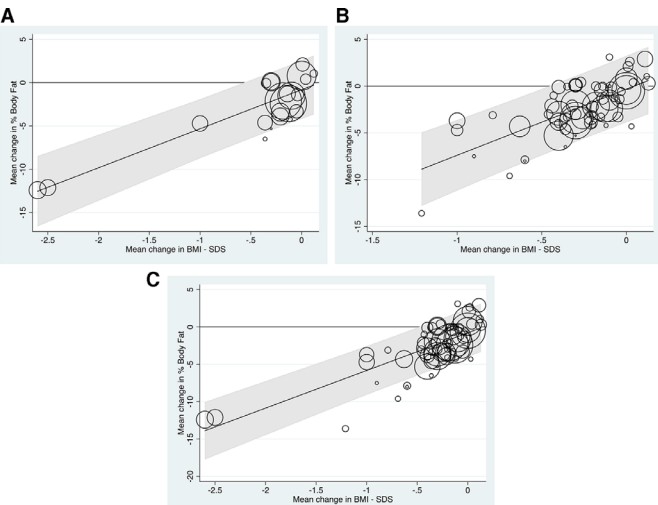

**Figure 5** Sensitivity analysis. BMI-SDS, body mass index-SD score. (A) Analyses based on the 22 subsets where the SEs of the mean changes in percentage Body Fat were known (Fitted meta-regression line: Mean change in % body fat = 4.502 x Mean change in BMI-SDS – 0.810). (B) Analysis using all data subsets but excluding two extreme values (reduction of mean BMI-SDS of more than 1.5), leaving 64 subsets (Fitted meta-regression line: Mean change in % body fat = 7.078 x Mean change in BMI-SDS – 0.318). (C) Analysis using all 66 data subsets but using a correlation coefficient of 0.50, rather than 0.81, to estimate the SE of the mean change in % Body Fat for the 66–22=44 subsets where this was not available (Fitted regression line: Mean change in % body fat = 5.039 x Mean change in BMI-SDS - 0.783).

the included studies were considered to be of moderate to high quality according to the HTA quality assessment tool.[30] Despite there being a positive relationship between mean change in percentage body fat and mean change in BMI-SDS, our modelling suggested that, in order to be confident of effecting a mean loss in percentage body fat, any future study should aim to reduce the BMI-SDS by at least 0.6.

### Strengths and limitations
We believe that this is the first paper to attempt to bring together all studies that have reported both a change in BMI-SDS and changes in a marker of adiposity in the paediatric population with obesity. The systematic methods employed to identify the included studies were stringent, but it is possible that some relevant studies might have been missed. In addition, there was some variation in the reporting of results where there were multiple publications of the same study; in these cases, the results from the most comprehensive paper have been used. An important limitation to address in the broader context going forward is whether BMI-SDS is the best way to represent changes in BMI at extremes of body weight. The US Center for Disease Control cautioned the use of BMI-SDS in weight extremes in 2009.[40] Freedman *et al* have suggested that there are better measures of adiposity in severe obesity, such as percentage of 95th percentile BMI ($\%BMI^{p95}$) or distance in kg/m² from the 95th

percentile ($\Delta BMI^{p95}$).[41] Other groups have identified alternate methods when dealing with extremes of obesity such as BMI%[42] or percentage above IOTF-25.[43] Vanderwell *et al* have also suggested that BMI-SDS is only a weak to moderate predictor of percentage body fat in children, especially under 9 years of age.[44] Notwithstanding these cautions, we based this analysis on the data available to us which was almost entirely reported in terms of BMI-SDS and continues to be the case in most recent publications to date.

It has been suggested that the relationship between change in percentage body fat and change in BMI-SDS may differ between very young and older children.[45] Our inclusion criteria stipulated ages from 4 to 19 years. Most of the studies spanned a wide range of ages (table 1) and we did not have access to individual child data to facilitate stratification by age. Data from four subsets of children up to 10 years,[37 46 47] however, did not suggest a different relationship from the whole cohort (see online supplementary appendix 3).

### Agreements and disagreements with other research

Previous research has shown that an improvement in body composition and cardiometabolic risk can be achieved with a BMI-SDS reduction of ≥0.25 in adolescents with obesity, with greater benefits achieved when losing at least 0.5 BMI-SDS.[48]

In clinical practice, the degree of weight loss with lifestyle intervention is moderate and the success rate 2 years after onset of an intervention is low (<20% with a decrease in BMI-SDS <0.25).[49] There have been numerous reports of lifestyle-based weight management interventions for children with obesity, many documenting changes in BMI-SDS, but a recent meta-analysis has documented that while such changes may be statistically significant, they are unlikely to lead to clinical improvements in metabolic health.[50 51] To our knowledge, this is the first paper to establish the minimum change in BMI-SDS required to be certain of improving adiposity as percentage body fat for children and adolescents with obesity in clinical trials.

### Clinical implications

If reducing fat mass is the aim of weight management interventions, our analysis in this review demonstrates that BMI-SDS changes must be of an order seldom achieved in trials worldwide. From our model, to be confident about ensuring an improvement in mean body fat, one should aim to reduce mean BMI-SDS by at least 0.6. Figure 3 and sensitivity analysis 5B (figure 5) suggest that to reduce body fat by 5% requires a much larger BMI-SDS reduction, of the order of 1.3–1.5, although there was a paucity of data in this region.

### Recommendations for future research

While we are undertaking further analyses looking at key cardiovascular and metabolic outcomes in childhood obesity that may demonstrate improvements at lesser levels of BMI-SDS reduction, the evidence suggests that very few childhood weight management trials to date are likely to have improved percentage body fat and calls in to question their overall efficacy in terms of health improvement. That said, any trial demonstrating an improvement of the magnitude of 0.6 BMI-SDS might be termed successful with a likely reduction in fat mass. However, given the mounting evidence that BMI-SDS may not accurately reflect adiposity at extremes of obesity, it seems prudent for future trials to report additional indices of derived BMI values which may better reflect changes in actual adiposity. Which of the many measures suggested eventually establishes itself as the 'optimal' determinant at extremes of body mass is yet to be determined?

### CONCLUSIONS

Using our model, to predict any fat mass improvement when reporting a weight management trial outcome requires a BMI-SDS decrease of 0.6. When evaluating key outcomes for future weight management trials and services, this figure needs to be borne in mind by researchers, healthcare professionals and commissioners when assessing apparent success.

**Contributors** LB and RP provided substantial contributions to the conception and design of the study, designed the data extraction instrument, performed electronic database searches, data screening, extraction and quality assessment, coordinated and supervised data collection and drafted and revised the manuscript. JPHS provided a substantial contribution to the conception and design of the study, conducted data screening and interpretation and assisted with drafting and revision of the manuscript. LPH provided statistical expertise in relation to study design and conducted the data analyses and contributed to the drafting and revision of the manuscript. RM, AC and RB were involved in data acquisition and management. All authors approved the final manuscript as submitted and agree to be accountable for all aspects of the work.

**Funding** This study was supported by the NIHR Biomedical Research Centre at the University Hospitals Bristol NHS Foundation Trust and the University of Bristol.

**Competing interests** JPHS and LPH are authors on two studies included in the systematic review that this paper reports on.

**Patient consent for publication** Not required.

**Provenance and peer review** Not commissioned; externally peer reviewed.

**Data sharing statement** Dataset will be available from the Dryad repository (not yet set up so DOI currently unavailable).

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
