## [Reviewer comments · BMJ Open]

ARTICLE DETAILS

TITLE (PROVISIONAL)	What change in body mass index is associated with improvement in percentage body fat in childhood obesity? A meta-regression
AUTHORS	Birch, Laura; Perry, Rachel; Hunt, Linda; Matson, Rhys; Chong, Amanda; Beynon, Rhona; Shield, Julian

VERSION 1 – REVIEW

REVIEWER	Prof Petur B Juliusson MD, PhD University of Bergen, Norway
REVIEW RETURNED	31-Dec-2018

GENERAL COMMENTS	Birch L et al. What change in body mass index reduces body fat in childhood obesity: a meta-regression This is a systematic review of lifestyle interventions in children and adolescents with obesity. The aim is to see how much change in BMI z-scores is clinically relevant when compared to % body fat (“in the sense that the associated 95% prediction interval for change in mean percentage body fat was wholly negative number”, as written in the abstract). My comments: 1. Page 3, line 7 (and many other places in the ms!). BMI SDS or BMI z-scores? I suggest the authors decide what to use. In the literature, both are in use! However, using both in the same paper, and not consequently, is not making things easier...2. Page 6, line 2-8. The introduction points out important knowledge gap, which is that we need information about “beneficial change (...in BMI...) in relation to cardiometabolic risk factors, rather than just statistical significance”. Page 5, l 42-50 and page 6, line 45-47 addresses this further. However, this study is not aiming to fill this “knowledge gap” at all. The authors have two other paper in the pipeline that are more focused on this, but this particular paper is “only” looking at changes in BMI versus changes in “body fat”. Therefore, I presume the introduction could be simplified and made clearer with this in mind.3. BMI z-scores has, as very briefly touched on in the very end of the Discussion, clear limitations in this particular group of children with obesity. Further publications: https://www.ncbi.nlm.nih.gov/pubmed/19776142 and https://www.ncbi.nlm.nih.gov/pubmed/28992355 The BMI observations in children with severe obesity, lying above the LMS generated BMI references, are problematic as one cannot properly correct for age, sex or degree of obesity. Despite of this, BMI z-scores have been widely in use. However, because of these
---

	limitations, the use of BMI z-scores might not be the optimal parameter for use in future research. I suggest this should be made clearer in the manuscript. 4. Page 7, line 33. BMI “standards”. The usual term would be “references”. 5. Page 8, line 45. I would suggest this paper focuses on its contents; it is not a “protocol paper”... 6. Page 10, line 33. The total number of children included into this analysis is not mentioned. 7. Table 1. In this table, there is no information about by what means % body fat was measured. 8. Page 18, lines 45 and 56. Interventions with duration of only 15 days are very short! The condition is chronic! Sample size down to 8, that “n” is very low! 9. Page 22, line 12. I might have missed it, but the line “Note that “intervention” here could have been the “control” arm in some cases” reads strangely. Could this be rephrased – made clearer? 10. My final comment: The title states, “What change in body mass index reduces body fat in childhood obesity”. However, “body fat” in this paper is actually phrased in the abstract as “adiposity as percentage body fat” – and looking into material and methods, one finds out one is talking about % body fat (I presume measured by BMI or DEXA, however I do not find information about what methods is being used in the different studies) and waist circumference. Waist circumference is certainly not a measure of “percentage body fat”.
--	--

REVIEWER	Olli Saarela University of Toronto, Canada
REVIEW RETURNED	08-Jan-2019

GENERAL COMMENTS	The authors present a systematic review and a meta-analysis on the relationship between BMI z-score reduction and percentage reduction in body fat in different kinds of intervention studies on obesity among children and adolescents. 39 studies were identified that reported both measurements, and were assessed for quality. As expected, a meta-regression suggested significant positive relationship between BMI change and body fat change. Based on prediction intervals from the meta-regression model the authors report that, with 95% confidence, a study showing 0.6 standard deviation reduction in BMI would also show a reduction in body fat. I don't have major issues with the methods of the study (other than the one mentioned below), but I think that the motivation of the research question and implications of the results would require some further explanation, as detailed below. Major comments:  - The authors correctly interpret the result from the meta-regression, but since the interpretation is at study level rather than at individual level, it is unclear to this reviewer what, if any, implications this has for planning future intervention studies on obesity. Would not the new studies be sized to achieve a predetermined power to detect a clinically significant effect size? Furthermore, if, as the authors argue, body fat reduction is a more important outcome than BMI reduction, would not the new studies be planned directly in terms of a reduction in the former rather than the latter? - Please specify in more detail how the prediction intervals were calculated. These do not seem to consider the size (standard error) of the study for which the prediction is made. Any justification for
---

	this? Minor comments:  - It would be good to see some residual diagnostics for the meta-regression, in addition to the predicted random effects. - It appears that control and intervention arms of the RCTs were analyzed as separate studies. This could be stated more clearly.
--	--

VERSION 1 – AUTHOR RESPONSE

Reviewer: 1. Prof Petur B Juliusson MD, PhD Institution and Country: University of Bergen, Norway
This is a systematic review of lifestyle interventions in children and adolescents with obesity. The aim is to see how much change in BMI z-scores is clinically relevant when compared to % body fat (“in the sense that the associated 95% prediction interval for change in mean percentage body fat was wholly negative number”, as written in the abstract).

My comments:

1. Page 3, line 7 (and many other places in the ms!). BMI SDS or BMI z-scores? I suggest the authors decide what to use. In the literature, both are in use! However, using both in the same paper, and not consequently, is not making things easier...

Thank you for this helpful comment; as you point out both terms are in use in the literature and are both reported in the papers included in this meta-regression. However, to make things clearer we will refer only to BMI-SDS throughout this paper and have revised the manuscript accordingly to explain this (page 5).

2. Page 6, line 2-8. The introduction points out important knowledge gap, which is that we need information about “beneficial change (...in BMI...) in relation to cardiometabolic risk factors, rather than just statistical significance”. Page 5, l 42-50 and page 6, line 45-47 addresses this further. However, this study is not aiming to fill this “knowledge gap” at all. The authors have two other paper in the pipeline that are more focused on this, but this particular paper is “only” looking at changes in BMI versus changes in “body fat”. Therefore, I presume the introduction could be simplified and made clearer with this in mind.

We thank the reviewer for this observation which we agree with entirely. We have deleted most references to cardio-metabolic health and made clear in both the introduction and outcomes measures that this paper addresses adiposity and BMI-SDS changes alone (page 8).

3. BMI z-scores has, as very briefly touched on in the very end of the Discussion, clear limitations in this particular group of children with obesity. Further publications:
<https://www.ncbi.nlm.nih.gov/pubmed/19776142> and <https://www.ncbi.nlm.nih.gov/pubmed/28992355>.
The BMI observations in children with severe obesity, lying above the LMS generated BMI references, are problematic as one cannot properly correct for age, sex or degree of obesity. Despite of this, BMI z-scores have been widely in use. However, because of these limitations, the use of BMI z-scores might not be the optimal parameter for use in future research. I suggest this should be made clearer in the manuscript.

We agree with the reviewer that we only touched briefly on the central issue of whether BMI-SDS is the most appropriate tool for measuring changes at body mass extremes. We have thus extended our discussion to include further examples (four critically important papers) that demonstrate the problems

associated with SDS changes at obesity extremes in the 'strengths and limitations section' (page 27). However, we still feel this paper is of merit as the vast majority of papers, including many published in the last year, report BMI-SDS changes alone. We have also extended our observations in the 'recommendations for further research' section (page 28) although no suggested derived measure has yet to prove itself the 'new' BMI-SDS.

4. Page 7, line 33. BMI "standards". The usual term would be "references"

Thank you, we have revised this to "references" in the manuscript (page 7).

5. Page 8, line 45. I would suggest this paper focuses on its contents; it is not a "protocol paper"

We have simplified the wording on page 8 to only include outcomes variables of direct relevance to body fatness or adiposity. We have referred the reader to the published protocol paper for the complete list of metabolic health markers considered in the full systematic review (page 8).

6. Page 10, line 33. The total number of children included into this analysis is not mentioned.

Thank you for this comment. The numbers in each study are summarised in Table 1 together with the numbers analysed in the study subsets. We have added the total number of children analysed in the Quantitative Analysis section (n=2,618) (page 23).

7. Table 1. In this table, there is no information about by what means % body fat was measured.

Thank for you highlighting this. We have now included the methods of percentage body measurements where they were available. Please refer to Table 1; new column: Method of % body fat measurement.

8. Page 18, lines 45 and 56. Interventions with duration of only 15 days are very short! The condition is chronic! Sample size down to 8, that "n" is very low!

We agree; however, we included all relevant studies with intervention duration over 2 weeks as per our systematic review inclusion criteria in our published protocol paper (<https://doi.org/10.1186/s13643-016-0299-0>).

9. Page 22, line 12. I might have missed it, but the line "Note that "intervention" here could have been the "control" arm in some cases" reads strangely. Could this be rephrased – made clearer?

We have replaced the word 'intervention' by 'study' (i.e. pre vs post study) (page 23) and have included some further clarification of the study subgroups in the same paragraph (see also our response to Reviewer 2, below).

10. My final comment: The title states, "What change in body mass index reduces body fat in childhood obesity". However, "body fat" in this paper is actually phrased in the abstract as "adiposity as percentage body fat" – and looking into material and methods, one finds out one is talking about % body fat (I presume measured by BMI or DEXA, however I do not find information about what methods is being used in the different studies) and waist circumference. Waist circumference is certainly not a measure of "percentage body fat".

Thank you – we have revised the title to "What change in body mass index is associated with improvement in percentage body fat in childhood obesity? A meta-regression" and trust that this more accurately reflects the content of this paper.

Reviewer: 2

Reviewer Name: Olli Saarela

Institution and Country: University of Toronto, Canada Please state any competing interests or state 'None declared': None declared.

The authors present a systematic review and a meta-analysis on the relationship between BMI z-score reduction and percentage reduction in body fat in different kinds of intervention studies on obesity among children and adolescents. 39 studies were identified that reported both measurements, and were assessed for quality. As expected, a meta-regression suggested significant positive relationship between BMI change and body fat change. Based on prediction intervals from the meta-regression model the authors report that, with 95% confidence, a study showing 0.6 standard deviation reduction in BMI would also show a reduction in body fat. I don't have major issues with the methods of the study (other than the one mentioned below), but I think that the motivation of the research question and implications of the results would require some further explanation, as detailed below.

Major comments:

- The authors correctly interpret the result from the meta-regression, but since the interpretation is at study level rather than at individual level, it is unclear to this reviewer what, if any, implications this has for planning future intervention studies on obesity. Would not the new studies be sized to achieve a predetermined power to detect a clinically significant effect size? Furthermore, if, as the authors argue, body fat reduction is a more important outcome than BMI reduction, would not the new studies be planned directly in terms of a reduction in the former rather than the latter?

We understand the reviewer's comment and it is true that you could design studies to improve measures of adiposity rather than BMI and its derived values. However, the vast majority of weight management interventions in childhood are community based and do not have access to relatively complex measuring devices that quantify body fat directly. Given that many studies would not have access to adiposity measurement devices directly, we believe that they could be powered for an intervention to achieve a BMI-SDS reduction of >0.6 and in this way they would be 'certain' to have the clinically meaningful result of lowering adiposity (percentage Body fat) which we believe to be vitally important in weight management.

Please specify in more detail how the prediction intervals were calculated. These do not seem to consider the size (standard error) of the study for which the prediction is made. Any justification for this?

Standard errors (SEs) of the mean changes in % body fat are taken into account in the analysis. (SEs of the mean changes in the predictor (BMI-SDS) however are not, but this is exactly analogous to ordinary linear regression in which the predictor variable is assumed to be known precisely.)

The sizes of the circles in the plots illustrate the precision of the estimates ($1/SE^2$), as would be used for a fixed effect model.

We did, however, use a random effects model, as fully described in the referenced paper (31: Harbord & Higgins, 2008) and implemented by them in Stata. Briefly (using their notation), each study subset provided an estimate of y_i , the mean change in %Body Fat, together with the (known) standard error for that estimate σ_i , (the SE above) associated with its corresponding mean change in BMI SDS, x_i . The methodology assumes true effect y_i is assumed to follow a Normal distribution about the linear predictor, ie $y_i = x_i \beta + u_i + \epsilon_i$ with u_i representing the study subset effect, distributed as $N(0, \tau^2)$, with τ^2 the between-study variance, and ϵ_i distributed as $N(0, \sigma_i^2)$. REML was used to estimate τ^2 and the coefficients β are estimated using weighted least squares with weights $1/(\sigma_i^2 + \tau^2)$.

We chose the prediction interval for the regression line, rather than a confidence interval because, it indicated the uncertainty about the true effect in a future study given a particular mean change in BMI SDS; the prediction interval uses the standard error of the forecast, stdf , which from the program documentation is $\sqrt{(\text{stdp}^2 + \tau^2)}$, where 'stdp' is the standard error of the prediction (the standard error of the fitted values excluding the random effects).

It would be good to see some residual diagnostics for the meta-regression, in addition to the predicted random effects.

It was unclear to us as to which particular diagnostics would be helpful here. It was important to know that the relationship was linear and that the estimated random effects were approximately Normal with no gross outliers. This we did. The only other option was to include empirical Bayes estimates (predictions including random effects). Fig 3 is repeated below with these shown as triangles; we decided not to include them in the paper.

- It appears that control and intervention arms of the RCTs were analyzed as separate studies. This could be stated more clearly.

Please see all the responses to the first reviewer (above). We have tried to clarify this in the Quantitative Analysis section.

As we state in the Analysis section of the Methods section (page 9), we were not trying to assess the relative effects of the various interventions, but rather to examine the relationship between the two outcomes (mean change in % body fat and mean change in BMI-SDS) across the range of study subsets. To this aim, we included all subgroups within the same study, however subdivided (i.e. intervention vs control, boys vs girls or good responders vs poor responders), using these in preference to aggregated results from the whole study where both were reported. Random effects meta-regression was used to allow for residual heterogeneity (between subset variation in the mean change in % body fat unexplained by mean change in BMI-SDS).

VERSION 2 – REVIEW

REVIEWER	Petur Benedikt Juliusson University of Bergen, Norway
REVIEW RETURNED	01-May-2019

GENERAL COMMENTS	The authors have adequately responded to my comments.
---

REVIEWER	Olli Saarela University of Toronto, Canada
REVIEW RETURNED	05-May-2019

GENERAL COMMENTS	No further comments.
----------------------